# WntA expression and wing transcriptomics illuminate the evolution of stripe patterns in skipper butterflies

Jasmine D. Alqassar[1], Teomie S. Rivera-Miranda[2], Joseph J. Hanly[1,3], Christopher R. Day[1,4], Silvia M. Planas Soto-Navarro[2,5], Paul B. Frandsen[6], Riccardo Papa[2,5,7,*] and Arnaud Martin[1,*]

## ABSTRACT

Skippers (Hesperiidae) form a distinct lineage of butterflies where the developmental mechanisms of color patterning have seldom been studied. Skipper wing patterns often consist of median stripes, and studies from the mid-twentieth century suggested these elements are homologous to the central symmetry system (CSS) found in nymphalid butterflies. Here we examined the expression of the signaling ligand gene *WntA*, known to mark the presumptive CSS patterns in nymphalids, in the silver-spotted skipper *Epargyreus clarus*, and found support for the homology of the CSS across 95 MY of evolutionary divergence. We generated an annotated genome for *E. clarus* and used RNAseq to profile gene expression along the wing proximo-distal (P-D) axis. These data suggest that the transcription factor genes *lobe*, *u-shaped*, and *odd-paired* are expressed in restricted P-D sections of the wing similarly to *WntA*, indicating potential roles in CSS patterning. In addition, developmental genes involved in wing P-D patterning in *Drosophila* – *dachsous*, *four-jointed*, *homothorax*, *tiptop/teashirt*, *vestigial*, *scalloped* – reveal similar expressions between Diptera and Lepidoptera on the wing P-D axis, suggesting a deep conservation of P-D patterning in insect wings. This work expands our understanding of the mechanisms shaping wing pattern evolution in butterflies.

KEY WORDS: Evo-devo, Pattern formation, Wing epithelium, WntA signaling, Hesperiidae, Nymphalid ground plan, Developmental homology, Proximo-distal patterning

## INTRODUCTION

How do butterflies make color patterns on their wings, and to what extent can these various stripes and spots be considered homologous across large phylogenetic distances? The Hesperiidae family includes

[1]Department of Biological Sciences, The George Washington University, Washington, 20052, DC, USA. [2]Department of Biology, University of Puerto Rico-Río Piedras, 00931, San Juan, Puerto Rico. [3]Smithsonian Tropical Research Institute, Gamboa, 0843-03092, Panama. [4]Molecular and Cellular Biology Laboratory, National Institute of Environmental Health Sciences, Durham, 27709, NC, USA. [5]Molecular Sciences and Research Center, University of Puerto Rico-Río Piedras, San Juan, 00931, Puerto Rico. [6]Department of Plant and Wildlife Sciences, Brigham Young University, Provo, UT, 84602, USA. [7]Dipartimento di Scienze Chimiche della Vita e della Sostenibilità Ambientale, Università di Parma, Parma, 43124, Italy.

*Authors for correspondence (riccardo.papa@upr.edu; arnaud@gwu.edu)

J.D.A., 0009-0008-4688-6704; T.S.R.-M., 0009-0004-6311-4600; J.J.H., 0000-0002-9459-9776; C.R.D., 0000-0002-1841-950X; S.M.P.S.-R., 0000-0002-8614-7454; P.B.F., 0000-0002-4801-7579; R.P., 0000-0002-7986-9993; A.M., 0000-0002-5980-2249

at least 3800 species of skipper butterflies (Bridges, 1994; Van Nieukerken et al., 2011) and is uniquely situated to help answer this question. Hesperiidae diverged about 95 MYA from the core lineage that encompasses the main butterfly families (Kawahara et al., 2023), including Nymphalidae, which represents the main model systems for studying developmental evolution. Skippers show dazzling variation in their wing patterns, and while the interest in their diversification was recently reignited by phylogenetic and population genomics studies (Li et al., 2019; Toussaint et al., 2018; Toussaint et al., 2025), few studies have investigated its developmental basis.

Deciphering the degrees of homology between butterfly wing patterns is crucial to understanding whether conserved processes and mechanisms potentiate their diversification. In the 1920s, comparative morphologists Süffert and Schwanwitsch described the wings of Nymphalidae as deriving from pattern elements known as symmetry systems, that together, form a homology system known as the nymphalid ground plan (Nijhout, 1978, 1991; Schwanwitsch, 1924; Süffert, 1927). Following the terminology of Schwanwitsch (Fig. 1A,B), the central symmetry system (CSS) is a prominent feature of this ground plan, taking the form of a color field in the median region of the wing, and delimited on each side by proximal and distal boundaries (called $M_1$ and $M_2$). In its simplest form, the CSS corresponds to a median stripe that runs from the anterior to posterior edges of the wing. But as it is one of the tenets of wing pattern diversification, it has taken a myriad of forms that make it difficult to identify. Attempting to understand wing patterns as variations of a few common themes, Schwanwitsch spent most of his career building a theory of pattern homology across Lepidoptera that extends the existence of putative CSS homologues throughout the entire order (Schwanwitsch, 1956). For instance, this model proposes that Hesperiidae show a CSS element similar to the one found in Nymphalidae, as illustrated by Schwanwitsch in the skipper *Pyrgus sidae* (Fig. 1C). It is worth noting that this prediction denotes two properties of the *P. sidae* CSS: (1) instead of showing a continuous stripe, forewings show a dislocation of the CSS, precisely where the $Cu_1$ vein occurs; (2) in hindwings, the CSS is often white or reflective, unlike in the forewings. This model makes homology calls that are counterintuitive, because this results in CSS elements that can shift in color within the same individual; in *P. sidae*, the CSS is black and gray in the forewing, and white in the hindwing. This dual activity – the ability of the CSS to manifest itself as patterns of different colors between forewings and hindwings – prefigures that ground plan elements are not mechanistically tied to a color fate. From a mechanistic point of view, it can thus be useful to think of color pattern formation as a two-step process first involving spatial patterning, and then color specification.

Gene expression provides important clues for probing homology relationships in developmental processes (DiFrisco et al., 2020; Pantalacci and Sémon, 2015). The gene *WntA* has emerged as a

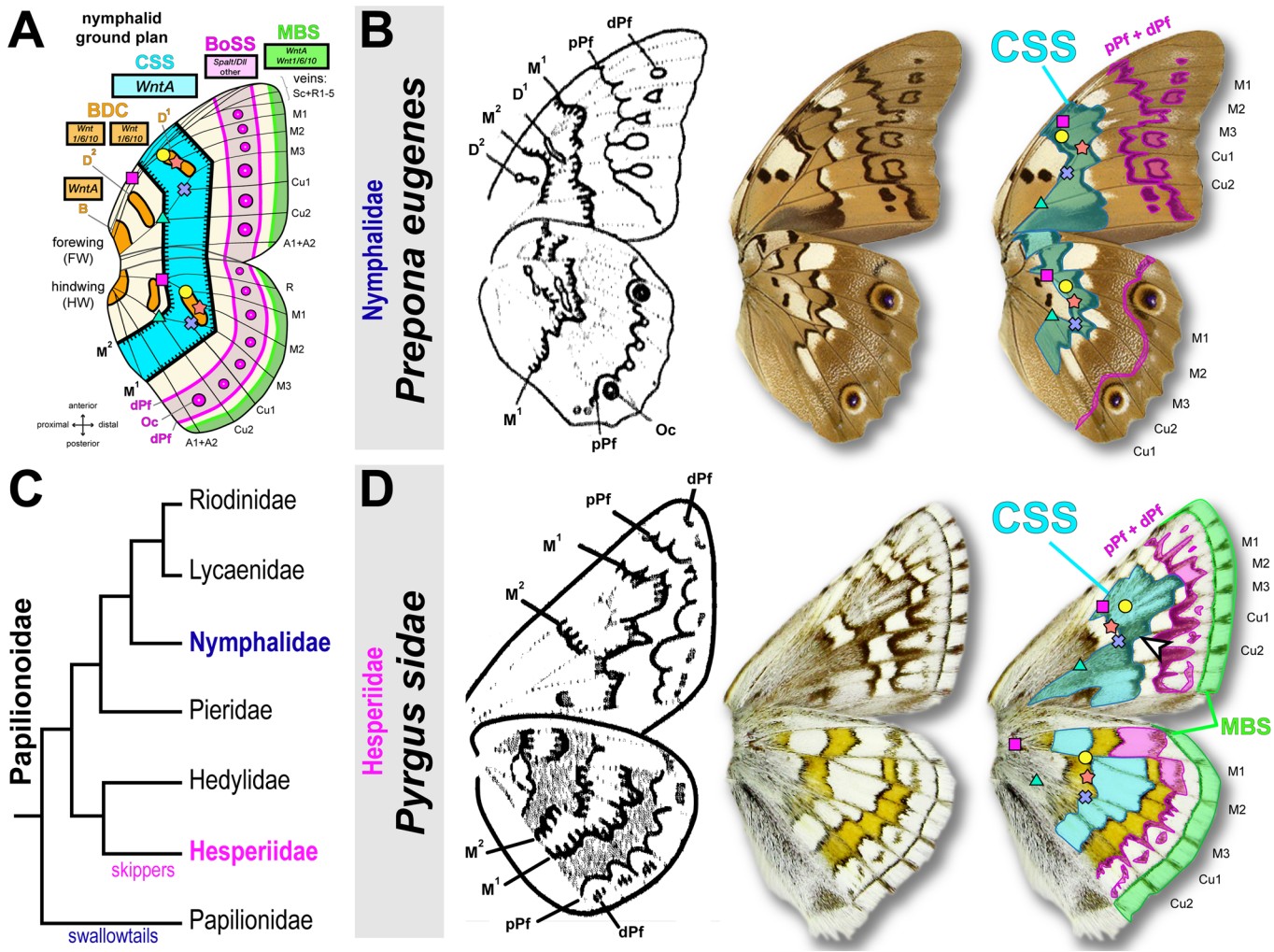

**Fig. 1. Hypotheses of stripe pattern homology between nymphalids and hesperiids.** (A) Current summary of the nymphalid ground plan, mainly based on the terminology of Schwanwitsch (Nijhout, 1991; Schwanwitsch, 1924) and including recent updates by Otaki and Mazo-Vargas et al. (Mazo-Vargas et al., 2017; Otaki, 2012; Otaki, 2021; Schwanwitsch, 1956). Discalis elements ($D_1$ and $D_2$); CSS, central symmetry system (cyan); BoSS, border ocelli symmetry system (magenta); Oc, forewing border ocelli; pPf and dPf, proximal and distal parafocal elements; MBS, marginal band system (green). Colored vignettes denote vein intersection landmarks. Magenta square, junction between R and M vein trunks; yellow dot, $M_1$-$M_2$ junction; red star, junction between discal crossvein and $M_3$; blue cross, $M_3$-$Cu_2$ junction; green triangle, $Cu_1$-$Cu_2$ junction. Rectangles feature the name of marker genes. (B) Ventral wing patterns of the nymphalid *Prepona eugenes* with ground plan annotations proposed by Schwanwitsch, 1956 (left: reproduction of published drawings; right: equivalent annotations as color overlays). (C) Phylogenetic relationship between Papilionoidae families. (D) Ventral wing patterns of the hesperiid *P. sidae* annotated as in panel A, and highlighting the inferred CSS predicted by Schwanwitsch, 1956 (left panel). According to this author, the CSS marks a grey pattern in forewings, and a dislocated white stripe pattern in hindwings, suggesting uncoupling of pattern and color state in fore/hindwings in skippers. The forewing CSS is markedly dislocated along the $Cu_1$ vein (arrowhead).

reliable spatial marker of the CSS and other patterns in nymphalid developing wing tissues (Concha et al., 2019; Gallant et al., 2014; Martin and Courtier-Orgogozo, 2017; Martin and Reed, 2014). *WntA* encodes a secreted ligand of the Wnt family that requires the Frizzled2 receptor to induce patterns of the basal, central, and marginal patterning systems (Banerjee et al., 2023; Hanly et al., 2021, 2023; Mazo-Vargas et al., 2017, 2022). WntA signaling has been understudied because this gene was lost both in *Drosophila* and in the vertebrate lineage, and little is known about its modes of signal transduction and transcriptional regulation. Most important, *WntA* shows a staggering diversity of expression patterns, and each of them prefigure adult color patterns of distinct colors across all nymphalid species assessed so far. WntA has thus emerged as a key signal that instructs spatial pattern formation during early pupal development in Nymphalidae, but with little data outside of this

family so far, except in Pieridae and Papilionidae (Fenner et al., 2020; Mazo-Vargas et al., 2024 preprint).

Here we selected the silver-spotted skipper *E. clarus* as a model system for the study of hesperiid wing pattern, as this North American skipper is easily collected on invasive kudzu patches, where larvae inhabit characteristic leaf shelters (Lind et al., 2001). We generated a genome assembly for this species and profiled the spatial expression of *WntA* to test for the homologies of the hesperiid and nymphalid CSS patterns. Then, we used RNAseq of pupal wing sections to conduct differential gene expression analyses comparing the transcriptomes of proximal, medial, and distal regions of pupal wings. We discuss how *WntA* expression supports the long-standing pattern homology system proposed by Schwanwitsch and detect a few candidate transcription factors that may be involved in the proximo-distal patterning of these systems in skippers and beyond.

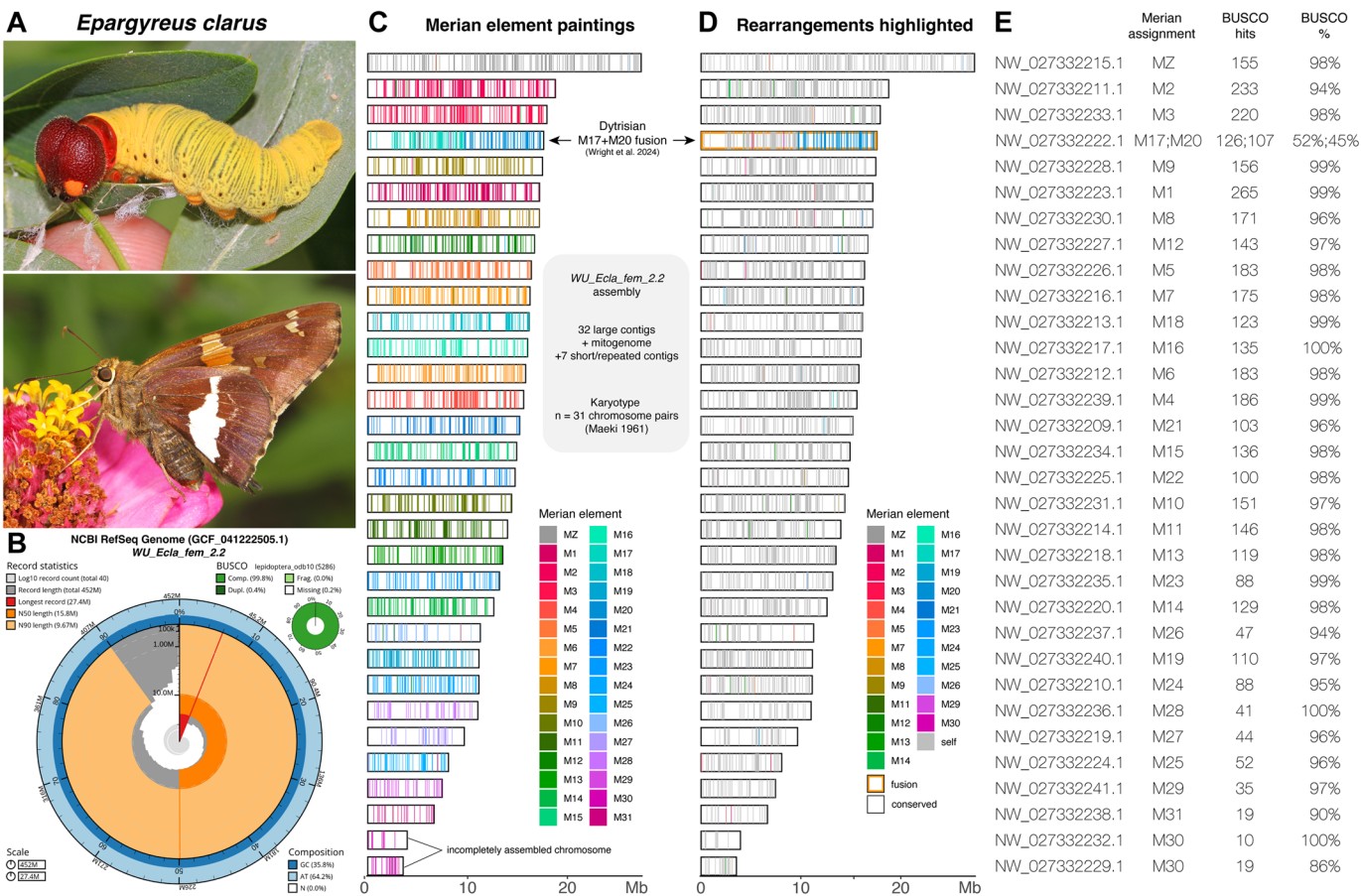

**Fig. 2. Assignment of the *E. clarus* genome to ancestral Merian elements.** (A) Fifth instar larva and adult form of the silver-spotted skipper *E. clarus* (photo credits, Judy Gallagher, CC-BY-2.0 license). (B) Summary SnailPlot assembly and completeness metrics for the *WU_Ecla_fem_2.2* RefSeq genome assembly of *E. clarus* (Challis et al., 2020). (C,D) Merian elements mapped across chromosomes in the *WU_Ecla_fem_2.2* haploid assembly of *E. clarus*. *N*=32 contigs are drawn to scale. BUSCO paintings with orthologue positions are shown as coloured bars, each shaded according to its corresponding Merian element (C), or with discrepancies with the ancestral lepidopteran linkage groups highlighted (D). The M17-M20 fusion is ancestral to Dytrisia, a lineage that encompasses most of lepidopteran diversity (Kawahara et al., 2019; Wright et al., 2024). (E) Merian assignments for each *E. clarus* contigs.

## RESULTS

### A reference genome for Hesperiidae

We generated a reference genome (Fig. 2A,B) and transcriptomes for *E. clarus* to enable future studies of Hesperiidae. High-molecular weight DNA from a female pupa was sequenced with PacBio HiFi reads, and assembled into a haplotype of 40 contigs and a total genome size of 452 Mb (N50=15.8 Mb; BUSCO analysis scores: complete single-copy: 99.4%, complete duplicated: 0.4%, fragmented: 0.00%, missing: 0.2%), including a complete mitogenome of 30.6 kb (NW_027332243.1). Assignments of the 32 main nuclear contigs to Merian elements (Fig. 2C,D), a system of homologous linkage groups conserved across Lepidoptera, recovered the set of 31 chromosomes ancestral to the Ditrysia lineage (Wright et al., 2024), implying the absence of any major chromosomal rearrangement in this species for about 150 MY (Kawahara et al., 2019). Two contigs are splitting the M30 element, but are unlikely to represent a real fission event, as they likely account for the discrepancy between our contig count (*n*=32) and the expected count of *n*=31 chromosomes established from the *E. clarus* karyotype (Maeki, 1961).

Transcriptomes include larval and pupal wings, whole heads from immatures and adults, ovaries, testes, and silk glands, and were generated using poly-dT cDNA synthesis (mRNA sequencing). Two samples were also prepared for total RNA sequencing using a ribosome depletion kit (QIAseq FastSelect –rRNA Fly Kit). This depletion method failed, resulting in over-representation of rRNA, but nonetheless includes non-coding transcripts that may increase transcript diversity.

A genome annotation was generated by the NCBI Eukaryotic Genome Annotation Pipeline team, and encompasses 15,809 predicted gene models, including 13,193 protein-coding genes. This high quality genome assembly and annotation for *E. clarus* are available online at the NCBI Datasets repository (*WU_Ecla_fem_2.2*, GCF_041222505.1) and represent the first NCBI RefSeq genome for Hesperiidae, a diverse family that includes more an estimated 3800-4100 skipper species (Bridges, 1994; Van Nieukerken et al., 2011). The *E. clarus* genome encodes the eight Wnt ligand genes previously found in other lepidopteran genomes (Ding et al., 2019; Fenner et al., 2020; Hanly et al., 2023; Holzem et al., 2019), including *WntA* and the conserved *Wnt1-Wnt9-Wnt6-Wnt10* cluster (Fig. S1). Because skippers occupy a key phylogenetic position (Fig. 1C) between the core group of butterflies and the early-diverging swallowtail lineage (Espeland et al., 2018; Kawahara et al., 2023), this reference genome fills a gap for comparative genomics in butterflies.

### *WntA* expression and heparin injections unveil the hesperiid CSS

We profiled the spatial expression of *WntA* mRNA using colorimetric *in situ* hybridizations in the fifth instar larval wing

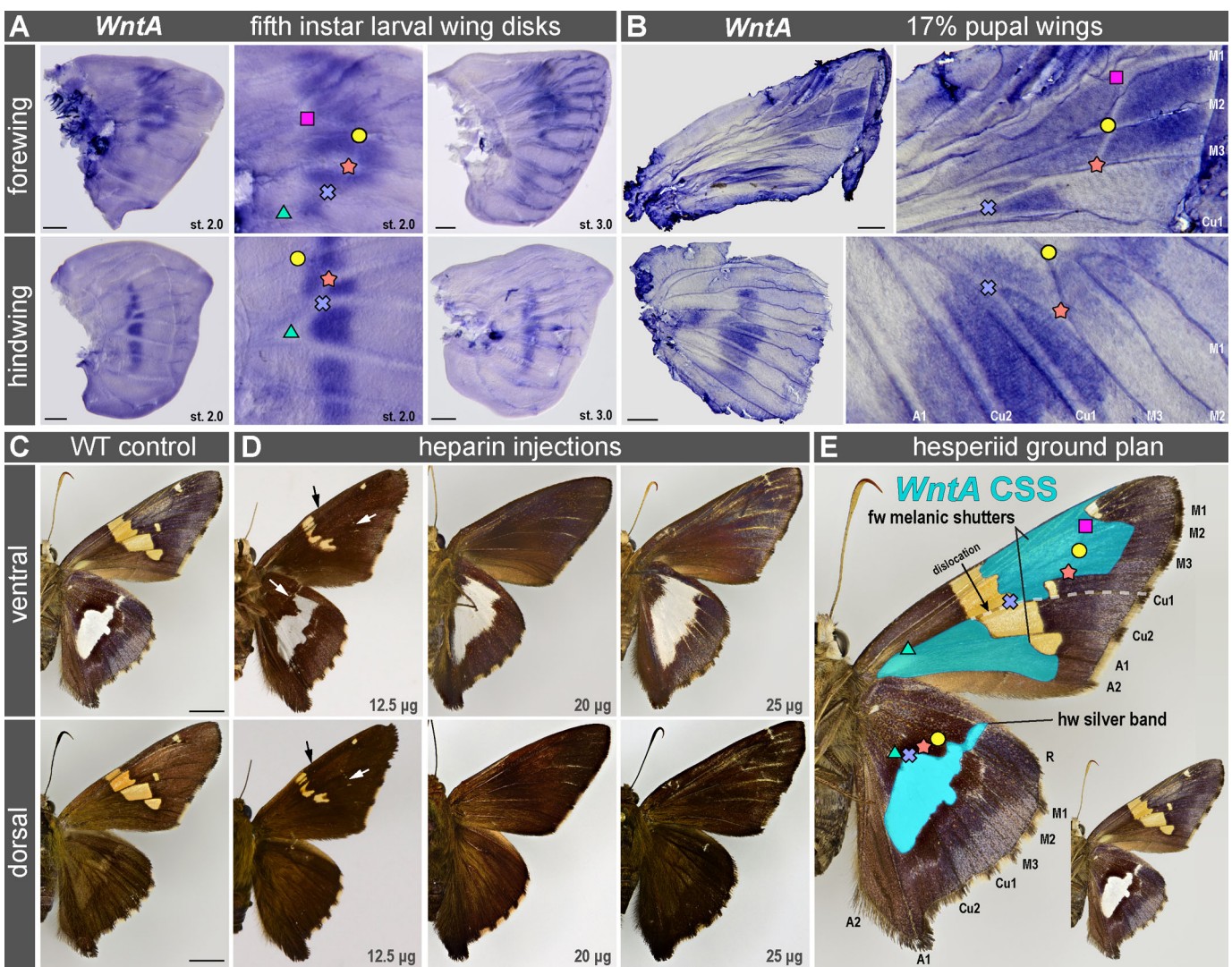

**Fig. 3. *WntA* marks CSS patterns with phenotypically dual effects in *E. clarus*.** Landmarks denote the point of bifurcation of presumptive veins (here, tracheal tube lumens) as in Fig. 1A. (A) *WntA* mRNA *in situ* hybridization in fifth instar wing disks. Stages 2.0 and 3.0 are based on nymphalid equivalent stages (Reed et al., 2007). Scale bars: 200 μm. (B) *WntA* mRNA *in situ* hybridization in wings at the 17% pupal stage. Scale bars: 1 mm. (C) Adult patterns in a wild-type control individual. Scale bars: 5 mm. (D) Heparin-induced pattern modifications, including the reduction and loss of the forewing orange band, and the expansion of the hindwing silver band. Arrows, areas of pattern expansion (melanic 'shutter' expansions in forewings, silver band expansion in hindwings) visible after intermediate doses of heparin injection. Effects were replicated across several individuals: 5/5 (12.5 μg), 1/3 (20 μg), 2/2 (25 μg). (E) Landmarking and inferred projection of *WntA* expression on adult patterns, marking a dual CSS homolog that corresponds to melanic 'shutters' in the forewing that flank the orange band, and to the silver band in the hindwing (see text for details).

disks and wings at 17% pupal development, two stages where *WntA* marks presumptive color patterns in nymphalids (Banerjee et al., 2023; Concha et al., 2019; Gallant et al., 2014; Hanly et al., 2023; Huber et al., 2015; Martin and Reed, 2014; Martin et al., 2012; Mazo-Vargas et al., 2017). Similarly to nymphalids, *E. clarus WntA* is expressed in the medial region of the wing and prefigures the position of adult patterns (Fig. 3A,B). In ventral hindwings, *WntA* clearly prefigures the silver bands. In forewings, *WntA* flanks the orange bands and spots and is instead expressed in brown-melanic regions. For example, *WntA* expression forms a triangle at the base of the $M_3$-$Cu_1$ vein intersection (blue cross in Fig. 3), which corresponds to a melanic region in adults, and that is immediately posterior to a small orange dot. This stereotypical expression of *WntA* at the base of the $M_1$-$Cu_2$ wing compartment resembles the expression of WntA in *Heliconius* and *Limenitis* species; in these nymphalids, WntA determines the shape of light-colored medial

patterns by inducing melanic patterns that have been called 'shutters', by analogy with shutters than slide onto a window and restrict light (Gallant et al., 2014; Gilbert, 2003; Martin et al., 2012; Nijhout et al., 1990). To test if WntA functionally induces distinct patterns in forewings and hindwings, we initially attempted CRISPR knock-outs but failed to microinject *E. clarus* eggs, due to the unusual toughness of their chorion. Instead, we used heparin injections as previously used in pre-CRISPR studies (Martin and Reed, 2014; Serfas and Carroll, 2005). Doses of 10-30 μg of heparin, administered within 16 h after pupation, were shown to consistently induce WntA gain-of-function effects across nymphalids, likely because heparin mimics the heparan sulfate proteoglycans of the extracellular matrix that facilitate Wnt transport and uptake (Binari et al., 1997; Gallant et al., 2014; Greco et al., 2001; Mazo-Vargas et al., 2017, 2022; Sourakov, 2018; Sourakov and Shirai, 2020). Following injection in the abdomen of early pupae, the forewing orange bands were either

reduced at the lower dose (12.5 µg) or completely absent at the higher doses (20-25 µg), an effect that emulates heparin effects in *Heliconius* and *Limenitis* and indicates an expansion of melanic shutter patterns (Fig. 3C,D). Conversely, while the hindwing silver band showed a small expansion of its anterior section under lower doses, higher doses resulted in drastic expansion and fuzziness of this pattern (Fig. 3C,D). Despite the caveat that heparin injections may affect other signaling ligands, the combination of *WntA* expression assays and pharmacological perturbation indicate that WntA functions as a melanic shutter in forewings and as a silver band activator in hindwings (Fig. 3E). These distinct effects reveal the dual nature of the CSS homolog in the hesperiid *E. clarus*.

### Transcriptomics of proximo-distal patterning in early pupal wings

The expression of *WntA* in the presumptive CSS pre-pattern implies that upstream developmental factors are partitioning the wing into proximo-distal subdivisions. To gain preliminary insights on this process, we conducted an RNAseq analysis of pupal forewings and hindwings (Tables S1-S11), each sectioned into proximal, medial, and distal portions (Fig. 4A). While *WntA in situ* hybridizations were conducted at around 17% of pupal development, we sampled wing transcriptomes at 12% of pupal development instead, in order to better capture early signaling events relevant to spatial pattern formation. DESeq2 analysis identified 1480 differentially expressed genes (DEGs) between the forewing and hindwing tissues (Fig. 4B,C; adjusted $P<0.05$): 931 with a forewing enrichment, and 549 with a hindwing enrichment including the Hox gene *Ultrabithorax* (*Ubx*), required for providing hindwing identity (Matsuoka and Monteiro, 2022; Tendolkar et al., 2021, 2024; Van Belleghem et al., 2023; Weatherbee et al., 1998). The high-expression and over-representation of genes assigned to the extracellular matrix in the forewing transcriptome suggests that this tissue is more actively involved in chitin production than the hindwings at the 12% pupal stage (Figs S3 and S4).

Next, we investigated the set of genes showing a pattern of differential expression along the proximo-distal axis of each wing (P-D axis DEGs). A total of 265 P-D axis DEGs were identified (Fig. 4B; adjusted $P<0.05$), here as well we saw an over-representation of transcription factors and signaling pathway genes in GO enrichment analyses (Fig. 4D); this finding supports the idea that the regionalization of the wing epithelium involves spatial regulation of genes with a regulatory developmental function. To visualize their spatio-temporal expression patterns, we generated heatmaps for all 265 genes (Fig. S2) and for the subsets corresponding to transcription factors and signaling pathway genes as determined by GO term annotation (Fig. 4E,F), using RNAseq data from whole fifth instar wing discs and from proximal, medial, and distal sections of wings at the 12% and 16% pupal stages.

Expression profiling validates the enrichment of *WntA* expression in the medial region of the pupal hindwings where the silver stripe is patterned (Fig. 5A). As *u-shaped* (*ush*), *odd-paired* (*opa*), *lobe* (*L*) are upregulated in the early medial hindwing in a similar fashion, it will be interesting to test in the future the expression of these transcription factors is connected to WntA signaling in presumptive color patterns. The transcription factors *homothorax* (*hth*), *tiptop* (*tio*, corresponding to *tio/tsh* in *Drosophila*), *zinc finger homeodomain 2* (*zfh2*) show proximal expression bias (Fig. 5B), consistent with previous transcriptomic studies of *Heliconius* early pupal wings (Hanly et al., 2019; Hines et al., 2012), and with their established roles as specifiers of proximal regions in *Drosophila* wing disks (Ruiz-Losada et al., 2018; Terriente et al., 2008; Zirin and Mann, 2004).

Inversely, the transcription factor genes *rotund* (*rn*), *vestigial* (*vg*), *scalloped* (*sd*), *bifid/optomotor-blind* (*omb*), as well as the genes *fat* and *four-jointed* (*fj*) showed distal enrichment profiles (Fig. 5C).

## DISCUSSION

### How to identify CSS patterns in Hesperiidae?

Weaving relationships of homology between rapidly evolving patterns requires the identification of core mechanisms that have been maintained since the divergence of two lineages. The conserved expression of *WntA* in the medial region of the developing wings of nymphalids and hesperiids likely indicates homology of the CSS patterns between these two lineages that separated about 95 MYA (Fig. 6A). Schwanwitsch, through his lifelong work examining pattern variation, had correctly predicted the position of the CSS and its dual nature in *P. sidae* (Fig. 1D). Indeed, *WntA* expression and heparin injections in *E. clarus* validated several properties of the CSS that are counterintuitive: first, that the forewing CSS is dislocated and acting as a melanic shutter, i.e. framing the more narrow stripes and dots of lighter color (Fig. 6E); second that the CSS can match a pattern of a different color identity than in the forewing, namely a white pattern in *P. sidae* and a silver band in *E. clarus*.

These data highlight the evolutionary malleability of hesperiid wing patterns and generate testable hypotheses about CSS homology in this lineage. While the identification of this element remains challenging without visualizing *WntA* expression, we can propose initial clues on the identity of the CSS in species with medial patterns. In nearly all nymphalid and hesperiid species we assessed, *WntA* consistently marks the tissue immediately distal of the $Cu_1$-$M_3$ vein junction, which is landmarked as a blue cross here, and as a blue dot in our previous studies (Concha et al., 2019; Gallant et al., 2014; Hanly et al., 2023; Huber et al., 2015; Martin and Reed, 2014; Mazo-Vargas et al., 2017). As a rule of thumb, the proximal $Cu_1$-$M_3$ area (arrowheads in Fig. 6B,C) is a good indicator of the color output of the CSS in a given wing surface, and we provide putative examples of this principle in a few Hesperiidae. Based on this, we propose that forewings of the rapidly diverging Pyrrhopyginae subfamily show a complete dislocation of the CSS along the $Cu_1$ vein (Fig. 6D,E). In nymphalids, the CSS is typically expressed as a continuous antero-posterior stripe, but it also often visible in a dislocated configuration with a break or shift along the $Cu_1$ vein, a phenomenon known as pierellization (Nijhout, 1991; Otaki, 2021; Penz, 2017; Schwanwitsch, 1925). The observed breaks of the CSS we document in the forewings of at least two skipper lineages are reminiscent of this effect (Figs 3E and 6). Thus, the developmental mechanisms that fuel and constrain the evolution of wing patterns are similar between nymphalids, hesperiids, and beyond.

We also caution these principles should not be applied too dogmatically given that the dynamics of wing patterning remain poorly understood; as known exceptions among nymphalids, the Monarch *Danaus plexippus* and Gulf fritillary *Agraulis incarnata* (formerly *Agraulis vanillae*) both underwent drastic shifts from the archetype of *WntA* expression as an antero-posterior medial stripe (Mazo-Vargas et al., 2017, 2022). The breathtaking diversity and rapid radiations of hesperiid wing patterns, particularly in contexts of wing mimicry and convergence (Li et al., 2019; Zhang et al., 2019), suggest that similar principles are at play in skippers and that other patterning tools than *WntA* may fashion this complex morphospace.

### Deep similarities between skipper and *Drosophila* P-D patterning

RNAseq profiling or early pupal wing development across three P-D subdivisions confirmed that *homothorax* and *tio/tsh* mark

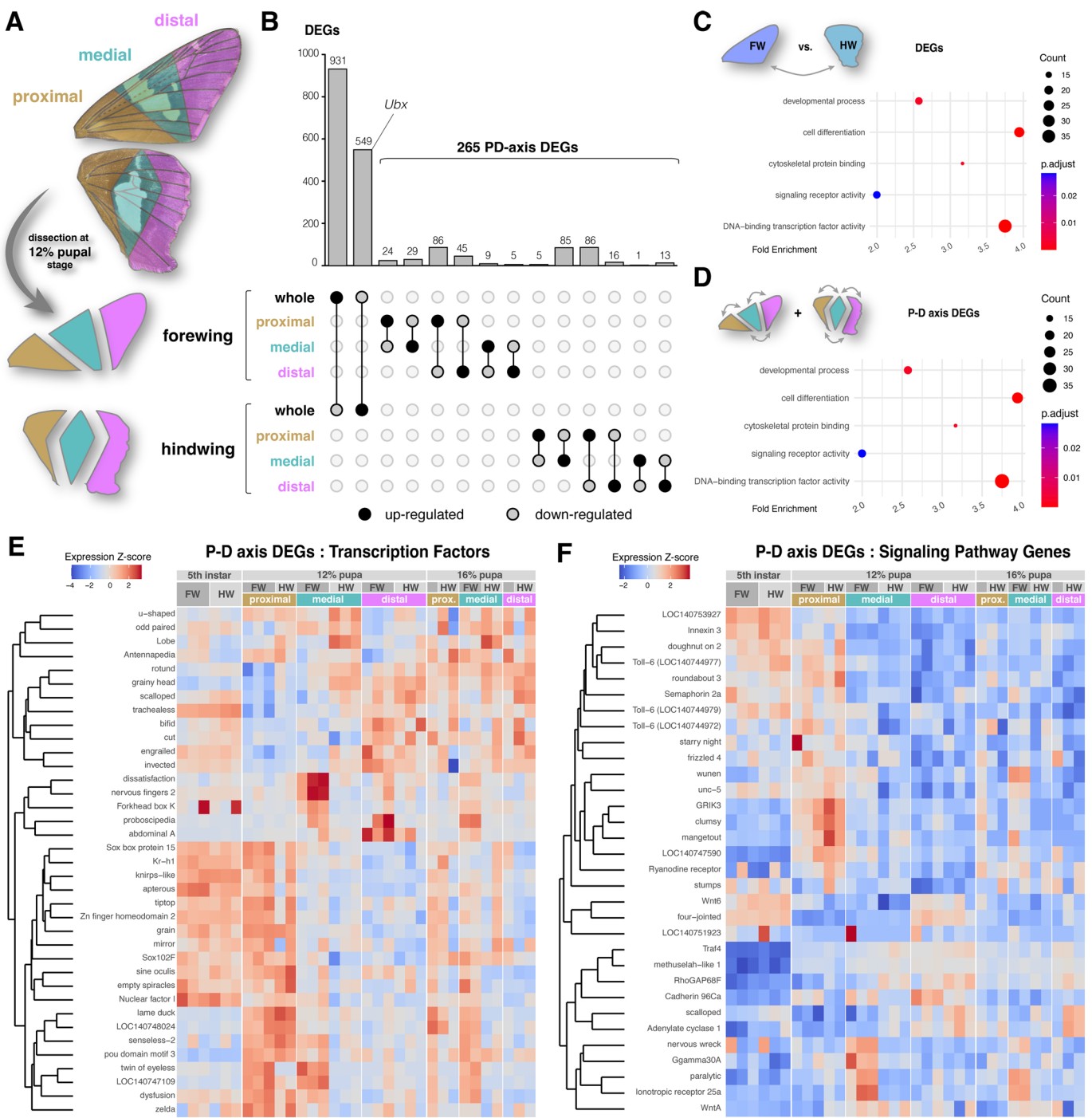

**Fig. 4. RNAseq and DEG analyses in *E. clarus* early pupal wings.** (A) Dissection of pupal wings along three proximo-distal sections, as projected on adult wings (top panel) to visualize the relative position of color pattern. Three biological replicates of each section were sequenced at the 12% pupal stage, except for proximal hindwings (two replicates). (B) Upset plot featuring the results of pairwise comparisons of differential expression at the 12% pupal stage between the forewing and hindwing, and all pairwise comparisons between sections within forewings and hindwings. The latter define a set of 265 proximo-distal axis DEGs. (C) GO enrichment analysis for reduced GO categories, featuring five GO categories over-represented in the FW versus HW DEGs. (D) The same analysis identifies four GO categories over-represented in the P-D axis DEGs. (E,F) Heatmap profiling of the 37 P-D axis DEGs classified as transcription factors (E), and signaling related genes (F). Columns each feature a different RNAseq sample. Forewing and hindwing wing disk samples dissected at the fifth instar larval stage, as well as 16% pupal stage wing samples sectioned as in panel A, are included to feature the expression of 12% stage DEGs over a wider temporal window.

proximal regions of the lepidopteran wing (Hanly et al., 2019; Hines et al., 2012), and this expression is reminiscent of their role in specifying proximal identity in the *Drosophila* wing disk (Zirin and Mann, 2004). Interestingly, *dachsous* showed a proximal pattern that is inverted compared to the distal-enrichment patterns of *fat* and *four-jointed*. Dachsous and Fat are interacting protocadherins whose opposing gradients, modulated by the kinase Four-jointed, establish the Fat-Dachsous planar cell polarity pathway that orients tissue polarity (Thomas and Strutt, 2012). In *Drosophila*, this system also regulates the growth of larval wing disks, with

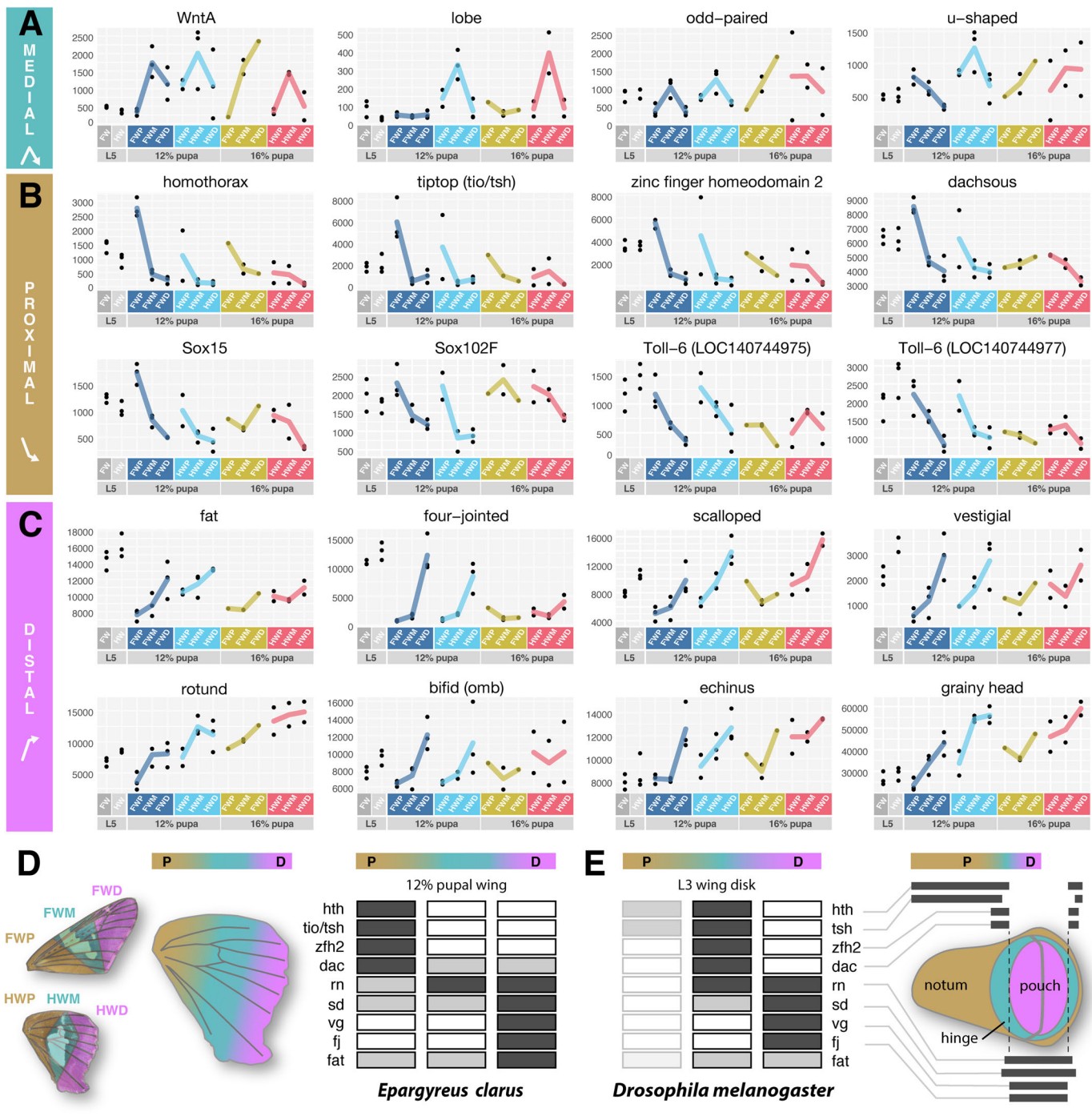

**Fig. 5. RNAseq profiling of selected patterning genes.** (A-C) Gene expression profiles for select DEGs, with normalized count values plotted on the y-axis. Panels A, B and C each group genes by expression profile in the 12% pupal stage hindwing. (A) Genes with enrichments in the median region, similar to *WntA*: *lobe*, *odd-paired*, *u-shaped*. (B) Genes with a proximal bias: *homothorax*, *tiptop*, *dachsous*, *zinc finger homeodomain 2* (*zfh2*), *Sox15*, *Sox102F*, and two tandem copies of *Toll-6* (LOC140744975 and LOC140744977). (C) Genes with a distal bias: *fat*, *four-jointed*, *rotund*, *vestigial*, *echinus*, *grainy head*, *scalloped*, *bifid* (*omb*). (D,E) Summary of the gene expression profiles of candidate proximo-distal specifiers in *E. clarus* pupal wings at the 12% pupal stage (D), and compared to the expression of their orthologues in the *D. melanogaster* wing imaginal disk (E), as described in previous publications (Cho and Irvine, 2004; Everetts et al., 2021).

*Dachsous* acting proximally in the body wall region, and Fat/Four-jointed acting more distally in the wing pouch, in conjunction with the Vestigial and Scalloped transcriptional co-factors (Cho and Irvine, 2004; Halder et al., 1998). Our finding that these genes are likewise enriched proximally (*dachsous*) or distally (*fat*, *four-jointed*, *vestigial*, *scalloped*) in skipper wings indicates that a P-D patterning module is deeply conserved. In spite of their distinct

wing imaginal disks, this reinforces the idea that P-D patterning in both flies and butterflies relies on a conserved set of transcription factors and signaling modules (Fig. 5D,E).

Our data also adds to the emerging evidence that this conservation of P-D patterning extends to other insect lineages such as Coleoptera and Orthoptera. Of note, both *sd* and *vg* show distal expressions in nymphalid butterflies (Banerjee et al., 2025; Carroll et al., 1994), and

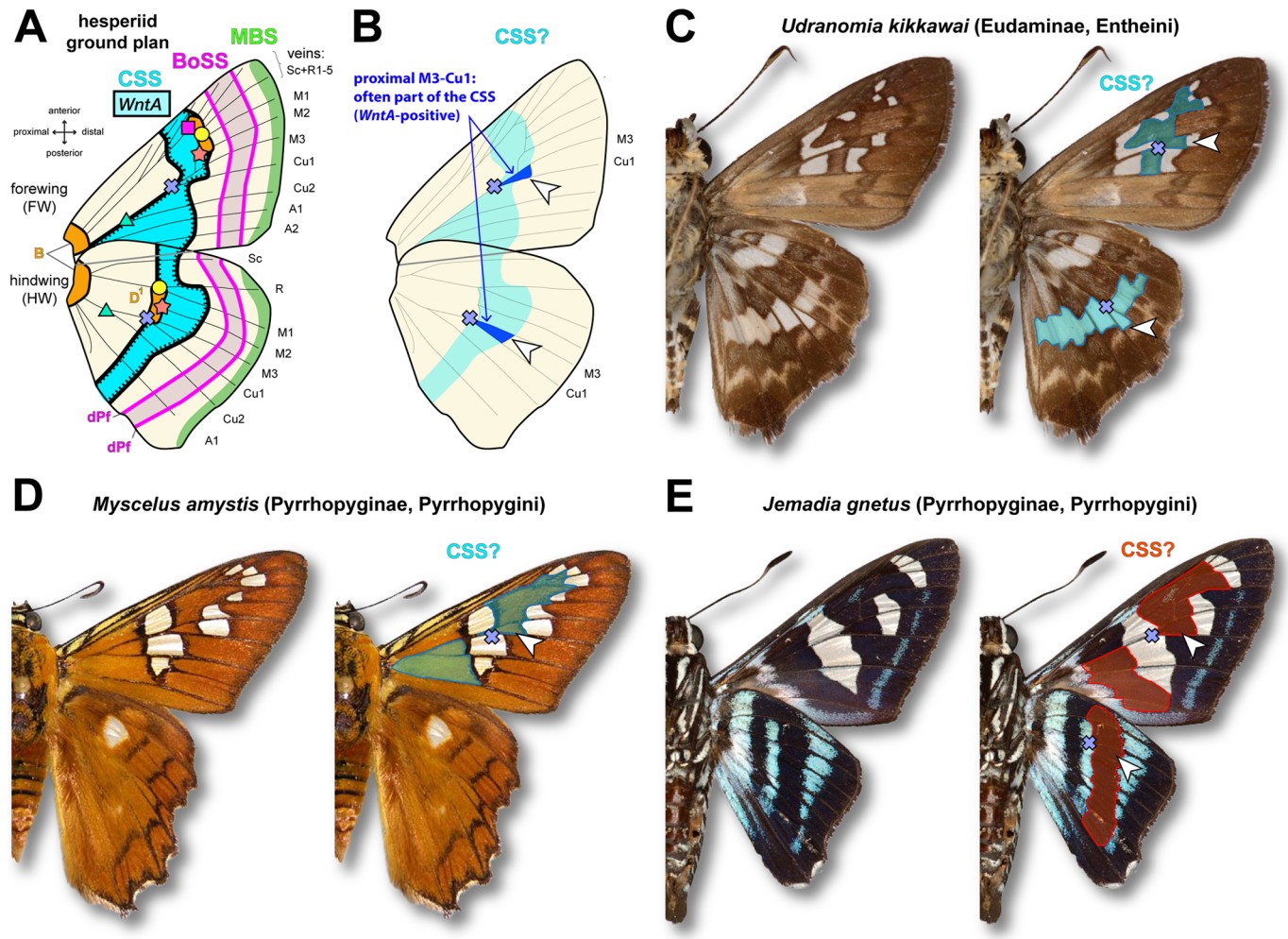

**Fig. 6. Extrapolation of the hesperiid ground plan to other species.** (A) Suggested hesperiid ground plan. Abbreviations and landmarks as in Fig. 1A. (B) Based on empirical observations in nymphalids, the pattern element (arrowhead) immediately distal to the $M_3$-$Cu_1$ vein junction (cross) can often be used to infer the identity of the CSS. (C) CSS inference in *Udranomia kikkawai*, a species from the same sub-family as *E. clarus* (Eudaminae), where a dual CSS is likely. (D,E) Examples of inferred CSS patterns in the rapidly diverging Pyrrohopyginae subfamily of firetip skippers. The forewings of these two species feature possible cases of 'pierellization', with a dislocation of the CSS along the $Cu_1$ vein, and with its distal boundary aligning with a basal patterning system in its anterior section. Photo credits: Nick V. Grishin (C,E) and Daniel H. Janzen (D), retrieved from the Butterflies of America online repository (https://www.butterfliesofamerica.com).

*sd* shows distal expression in the developing wings of crickets (Yamashita et al., 2023). A distal bias expression of *omb* was previously documented in butterflies, beetles and crickets (Banerjee and Monteiro, 2023 preprint; Ohde et al., 2022; Tomoyasu et al., 2009). And lastly, both *fat* and *fj* showed a distal enrichment in developing cricket wings (Ohde et al., 2022). More systematic studies of the expression and function of these genes are needed in various insect lineages, but the current data point at a deep homology of wing proximo-distal patterning across both hemimetabolous and holometabolous insects.

### New insights in the regulatory genetics of pupal wing development

While the development of wing tissues may follow conserved principles, the color patterns that decorate them also involved the evolution of novel gene expression, such as the pattern-specific domains of expression of *WntA* in Nymphalidae (Banerjee et al., 2023; Concha et al., 2019; Martin and Reed, 2014; Mazo-Vargas et al., 2017, 2022), Papilionidae (Mazo-Vargas et al., 2024 preprint), and in Hesperiidae. Little is known so far about the expression

of *u-shaped*, *odd-paired*, and *Lobe* in lepidopteran wings. *In situ* hybridization experiments in skippers and nymphalids will be particularly useful to determine whether they are associated with WntA patterning, and if their expression undergoes divergent or conserved evolution.

In addition to these medial signals (Fig. 5A), we also found enrichment profiles in the proximal region of the early pupal wing that were more pronounced, for the transcription factors *Sox102F*, *Sox15*, and tandem-paralogs of the Toll-6/Tollo receptor (Fig. 5B). Proximal enrichment was more pronounced in 12% wings, a stage the scale organ precursor cell lineage (SOP) has differentiated from other epithelial cells and are progressively undergoing a second round of cell division to give rise to scale and socket daughter cells (Loh et al., 2025a). As *Sox102F* marks epithelial (non-SOP) cells (Loh et al., 2025b), and as *Sox15* marks socket cells (Loh et al., 2025a,b; Prakash et al., 2024), their proximal enrichment in our 12% RNAseq samples could be thus due to a wave-like event of cell differentiation starting in the proximal region, as suggested by time-lapse observations of the early pupal wing epithelium (Iwata et al., 2014; Loh et al., 2025a). The function of *Toll-6* genes remains to be

determined, possibly hinting at a novel role of Toll-like receptor signaling during insect wing patterning.

## Conclusion

Overall, this work shows that developmental signal *WntA* shapes butterfly wing evolution across deep time and uncovers new candidate genes for color patterning. Moreover, the apparent similarities in wing spatial expression suggest that lepidopteran wing development may share deeply conserved mechanisms of proximo-distal patterning with *Drosophila*.

## MATERIALS AND METHODS

### Skipper butterflies

Immature stages of *E. clarus* (Cramer, 1775) were collected as larvae on kudzu vines around Washington, DC, USA, in the summers of 2018, 2019, and 2024. Larvae were provided kudzu cuttings fit into floral water vials in Tupperware containers and reared in a growth chamber with 40-60% relative humidity and a 14:10 h light:dark cycle, set at 26°C or 32°C (see below). Pre-pupae were transferred to transparent cups and pupation time was recorded using a TX-164 timelapse camera (Technaxx).

### Genome sequencing and annotation

High-molecular weight DNA was extracted from an *E. clarus* female pupa using a Qiagen Genomic-tip 100/g before preparation and sequencing of a PacBio HiFi library on a PacBio Revio sequencer at the Institute for Genome Science at the University of Maryland. Sequencing reads (NCBI SRA SAMN38816873) were assembled into a primary haplotype assembly using *Hifiasm* with a variety of -*s* values (from the default value of 0.55 to 0.35). The best assembly was chosen based on the highest contiguity and lowest duplication rate (assessed with *Compleasm*). Despite testing a variety of -*s* values, duplicate contigs remained in the assembly and were subsequently removed using *purge_dups* (Cheng et al., 2021; Guan et al., 2020). Genome assembly quality was assessed using *BUSCO v.6.0.0* and *Compleasm* v0.2.6 using the *Lepidoptera_odb10* lineage dataset and *BlobToolKit* (Challis et al., 2020; Huang and Li, 2023; Tegenfeldt et al., 2025). Painting of Merian elements onto GCF_041222505.1 scaffolds (Fig. 2C,D) was performed with *Lep_BUSCO_Painter* v.1.0.0, using BUSCO ortholog assignments from the reference set of lepidopteran ancestral linkage groups (Wright et al., 2024).

Non-wing tissue transcriptomes generated for annotation purposes – a whole third instar larvae, a fifth instar larval head, a pair of fifth instar silk glands, a male adult head, a female adult head, the testes from one adult male, and the ovaries from one adult female – were obtained by tissue storage in TRI Reagent (Zymo Research), followed by RNAseq library preparation and sequencing at a target depth of 30 M 150PE reads per sample by Azenta/Genewiz. These RNAseq sequencing reads are available under the NCBI BioProject PRJNA1165403. Genome annotation was performed by the NCBI Eukaryotic Genome Annotation Pipeline team; this resource is available as the *Epargyreus clarus WU_Ecla_fem_2.2* annotated reference genome (GCF_041222505.1). The GTF file produced from this annotation effort was manually curated for the purposes of this study to include the distant exon 1 of the gene *Cortex* from wing transcriptomic evidence generated from this study (Dataset S1). Additionally, the gene feature list was manually curated to include functional annotations, such as the *Drosophila melanogaster* genes with sequence similarity found by performing reciprocal BLASTp between the *WU_Ecla_fem_2.2* RefSeq_protein sequences and the translated protein sequences from the FB2025_02 FlyBase release of the *D. melanogaster* genome annotation using the BLAST+ tool (Camacho et al., 2009). The column *gene_label* was also added to the gene feature list to apply recognizable and shortened gene names from the NCBI and *D. melanogaster* functional annotations for clear data visualization. Lastly, annotation names for the Wnt ligand genes were manually assigned using a phylogenetic analysis (Fig. S1), based on a maximum likelihood phylogeny computed using W-IQ-tree default parameters, from an amino-acid sequence alignment produced by MAFFT in Geneious Prime and trimmed with and filtered with Guidance2 with a reliability threshold of 0.101 (Katoh and Standley, 2013; Sela et al., 2015),

which added predicted *E. clarus* Wnt proteins to a previously published set of Wnt protein sequences (Hanly et al., 2021).

### *In situ* hybridizations

An antisense riboprobe targeting *E. clarus WntA* (NCBI GenBank: XM_073090400.1) was PCR amplified from larval wing disk cDNA using the following primers (forward: 5′-CGAAGCAGCATTCGTACAC G-3′; *T7-promoter*+reverse:
   5′-*TAATACGACTCACTATAGGG*GGTAGCCTCTTCCACAGCAT-3′), transcribed with a Roche T7 DIG RNA labeling kit, purified with Ambion MEGAClear columns, and stored at −80°C. Detection of *WntA* mRNA expression in larval and pupal wings followed previously published procedures (Hanly et al., 2023; Martin and Reed, 2014), using 30 ng/ml of riboprobe during hybridization steps, and BM Purple (Roche) for the staining procedure. Pupal wings were dissected at 48-52 h after pupation at 26°C, corresponding to about 17% of pupal development (average total pupal development time at 26°C was measured as 292 h, *N*=8). Images were taken using a Nikon D5300 camera mounted to a Nikon SMZ800N trinocular dissecting microscope, equipped with a P-Plan Apo 1X/WF 0.105 NA 70 mm objective and a Nikon C-DS stand for diascopic mirror illumination.

### Heparin injections

Pupae between 3-14 h after pupa formation (32°C) were injected in their abdomen with 2.5-5 µl of 5 µg/µl heparin sodium salt (Sigma-Aldrich, H3393) or with water, using microcapillary needles fit into a Nanoject III microinjector. All treated pupae emerged into adult butterflies, which were pinned and imaged with a Nikon D5300 camera mounted with a Micro-Nikkor 105 mm f/2.8G lens.

### Pupal wing RNAseq

Wing transcriptomes of both male and female individuals are available under the NCBI BioProject PRJNA660444 and were prepared as follows, using a published procedure (Hanly et al., 2019). Larvae and pupae were reared at 26°C until dissection and storage in RNA later. Pupal wings were dissected at 36 h (12% pupal development) and 48 h after pupation in cold PBS (16%) and cut using microdissection scissors into six compartments: FWP (proximal forewing), FWM (medial forewing), FWD (distal forewing), HWP (proximal hindwing), HWM (medial hindwing), and HWD (distal hindwing). Each wing dissection used vein landmarks in order to result in similar sections across samples, as schematized in Fig. 4A. For RNA extraction, wing tissues were transferred into 2 ml tubes containing 500 µl of Trizol Reagent (Thermo Fisher Scientific) and homogenized with a Tissue Lyser for 2 min at 30 Hz. 200 µl of chloroform were added to the lysate before vigorous shaking for 15 s, settling for 3 min at room temperature, and centrifugation at 13,000 rpm for 15 min at 4°C. The aqueous top phase was transferred to a new tube, measured, and the same volume of 70% EtOH was added drop by drop to avoid localized precipitation. Total RNA was purified using the RNeasy kit (Qiagen), treated with DNASe I (Ambion) at 37°C for 10 min before addition of 5 µl of inactivation reagent, and stored at −80°C. Library preparation, and sequencing were performed by the Sequencing and Genomics Facility (University of Puerto Rico Rio Piedras, San Juan, Puerto Rico). Samples were sequenced at either 75 SE or 150 PE on an Illumina NextSeq 500 sequencer. Adapters were removed using Trimmomatic v0.39 using default parameters with the exception of the values SLIDINGWINDOW:4:20 and MINLEN:85 specified for PE samples (Bolger et al., 2014). Post-trimming, FastQC v.0.11.8 was used to verify all adapters were successfully removed and to assess read quality (https://github.com/s-andrews/FastQC). Reads were aligned to the *Epargyreus clarus WU_Ecla_fem_2.2* reference genome (GCF_041222505.1) with STAR v.2.7.11 (Dobin and Gingeras, 2016). STAR alignment was repeated with the IntronMotif output from the original alignment to better resolve splice junctions prior to read counting.

### Differential gene expression analyses

*FeatureCounts* (Subread v.2.0.8) was used to perform read counting (Liao et al., 2014) with the manually curated GTF annotation file (Dataset S1). This count dataset was used to perform differential expression analyses

using DESeq2 in RStudio (Love et al., 2014). First, we investigated which genes were differentially expressed between forewings and hindwings at 12% pupal development, by collapsing the counts from each compartment (proximal, medial, and distal) of each individual forewing or hindwing prior to count normalization. The experimental design ~ *wing_type* was then used to perform differential expression analysis with a significance threshold of adjusted *P*<0.05. The R package *ggplot2* (Wickham and Sievert, 2009) was used to produce Volcano and MA plots to visualize these data (Fig. S3).

Next, to define the set of P-D axis DEGs at 12% pupal development, we used the experimental design ~ *compartment*, resulting in six combinations of pairwise comparisons (three between the three forewings compartments, and three between the three hindwing compartments) with a significance threshold of adjusted *P*<0.05. To build a heatmap of z-score expression of genes across compartments and developmental timepoints, the experimental design ~ *compartment* was re-run on the entire count dataset to produce normalized counts for all samples, and used to build gene expression profile plots for genes of interest and a heatmap restricted to the set of P-D axis DEGs identified at the 12% pupal developmental stage using the R package *ggplot2* (Wickham and Sievert, 2009). Hierarchical clustering was applied to the dataset to cluster by similar gene expression profiles prior to visualization by performing variance stabilizing transformation using the function *vst()* from DESeq2, z-score scaling, and use of the R package *ggdendro* (https://andrie.github.io/ggdendro). The clustering *data* and dendrogram were then incorporated into the heatmap visualization produced by *ggplot2*. The *vst()* output was also used to run a principal component analysis with the *plotPCA()* function of DESeq2 (Fig. S4).

Gene Ontology (GO) Enrichment Analyses were performed separately on both the wing_type and compartment DEG datasets with *clusterProfiler* (Xu et al., 2024), using the *E. clarus* GO annotation hosted on the NCBI FTP server (*WU_Ecla_fem_2.2*), after it was mapped to the *slimGO_agr* and *slimGO_Drosophila* standardized subsets using *Map2Slim* in the OWLTools v4.5.29 package (https://github.com/owlcollab/owltools). Dot plots for each GO enrichment analysis were produced by *ggplot2* (Fig. 4C,D; Fig. S3). Transcription Factors were then defined from *slimGO_agr* re-mapped annotations as the subset of genes matching the molecular function categories GO:0003700 or GO:0008134 ('DNA-binding transcription factor activity', or 'transcription factor binding'). Signaling pathway genes were similarly defined as the subset of genes with a biological process annotation matching GO:0023052, GO:0038023, or GO:0005102 ('signaling', 'signaling receptor activity', or 'signaling receptor binding'). Filtered heatmaps with the corresponding gene category subsets with hierarchical clustering of gene expression profiles were then produced according to the aforementioned procedure (Fig. 3E,F).

## Acknowledgements
We thank Martha Weiss, Allison Brackley, Grace Jeschke, Anna Ren, Mariana Abarca, and John Lill for their support rearing *E. clarus* larvae, Ling Sheng Loh for help dissecting pupal wings, Yadira Ortíz for RNAseq library preparations, Anyi Mazo-Vargas for providing a DNA extraction, Daniel Davis for assisting with genome sequence submission, as well as Humberto Ortiz-Zuazaga and Rémi Megret for computational assistance. The annotation of the *E. clarus* genome was graciously carried out by the NCBI Eukaryotic Genome Annotation Pipeline team. We are also grateful for the Butterflies of America initiative to have made available an important photographic record of New World Hesperiidae.

## Competing interests
The authors declare no competing or financial interests.

## Author contributions
Conceptualization: J.J.H., R.P., A.M.; Formal analysis: J.D.A., T.S.R.-M., J.J.H., P.B.F.; Funding acquisition: R.P., A.M.; Investigation: J.D.A., T.S.R.-M., J.J.H., C.R.D., S.M.P.S.-N., P.B.F., A.M.; Supervision: R.P., A.M.; Writing – original draft: J.D.A., A.M.; Writing – review & editing: J.J.H.

## Funding
This work was funded by the National Science Foundation, Division of Integrative Organismal Systems grant (IOS-1656553) and a Division of Molecular and Cellular Biosciences grant (MCB-2217156) to A.M.; a Division of Molecular and Cellular Biosciences grant (MCB-2217155) to P.B.F.; Office of Integrative Activities grants (OIA-2435987 and OIA-1736026) to R.P. Personnel and equipment from the UPR-RP Sequencing and Genomics Facility was supported by the National Institute of General Medical Sciences award (P20GM103475). Open Access funding provided by The George Washington University. Deposited in PMC for immediate release.

## Data and resource availability
The *WU_Ecla_fem_2.2* NCBI RefSeq genome sequence and annotation for *E. clarus* is available on NCBI Datasets under the identifier GCF_041222505.1. The manually curated annotation GTF file and gene list derived from this resource can be found as Dataset 1 and Dataset 2, respectively. Wing transcriptomes are available on the NCBI SRA under NCBI BioProject PRJNA660444. The detailed results from all differential expression analyses performed in this study can be found in Tables S1-S11. All code associated with the differential expression analyses performed in this study can be found in a GitHub repository (https://github.com/jasalq/jasalq/Skipper_WntA_RNAseq).

## First Person
This article has an associated First Person interview with the first author of the paper.

## Peer review history
The peer review history is available online at: https://journals.biologists.com/bio/lookup/doi/10.1242/bio.062297.reviewer-comments.pdf

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
