## [Peer Review File · Biology Open]

WntA expression and wing transcriptomics illuminate the evolution of stripe patterns in skipper butterflies

Jasmine D. Alqassar, Teomie S. Rivera-Miranda, Joseph J. Hanly, Christopher R. Day, Silvia M. Planas Soto-Navarro, Paul B. Frandsen, Riccardo Papa and Arnaud Martin
DOI: 10.1242/bio.062297

Editor: Alissa Armstrong

Review timeline

Original submission:	1 October 2025
Editorial decision:	8 October 2025
First revision received:	23 October 2025
Accepted:	28 October 2025

Original submission

First decision letter

MS ID#: bio.062297

MS Title: WntA expression and wing transcriptomics illuminate the evolution of stripe patterns in skipper butterflies

Authors: Jasmine D. Alqassar, Teomie S. Rivera-Miranda, Joseph J. Hanly, Christopher R. Day, Silvia M. Planas Soto-Navarro, Paul B. Frandsen, Riccardo Papa and Arnaud Martin

I have now reached a decision on the above manuscript.

The reviewer reports are shown at the bottom of this email or can be accessed, together with a copy of this decision letter, by going to:

As you will see, the reviewers gave favourable reports, but raised some critical, overlapping points that will require amendments to your manuscript. Both reviewers raised an issue concerning accessibility of the manuscript to a broad readership. I hope that you will be able to carry out the suggested revisions, because we would like to be able to accept your paper.

At this stage, we also ask you to ensure your manuscript complies with our formatting guidelines - please see our manuscript preparation guidelines for details. Provided you are able to fully address the referees' comments, we are positive about publication of your paper (we accept over 95% of revision submissions) and therefore hope you won't mind any extra work involved in reformatting your manuscript at this point.

Please upload both a 'clean' version of your Word file, along with a highlighted version clearly showing where you have made changes in the revised manuscript. Please avoid using 'Track changes' in Word files as these are lost in PDF conversion.

I should be grateful if you would also provide a point-by-point response detailing how you have dealt with the points raised by the reviewers in the 'Response to Reviewers' box. Please attend to

© 2025. Published by The Company of Biologists under the terms of the Creative Commons Attribution License (<https://creativecommons.org/licenses/by/4.0/>).

all of the reviewers' comments. If you do not agree with any of their criticisms or suggestions please explain clearly why this is so.

Reviewer 1

Comments for the author

The study investigates wing pattern formation in Hesperidae, a butterfly family that has been largely understudied. By focusing on *Epargyreus clarus*, the authors provide rather compelling evidence for developmental homology between the Hesperidae and Nymphalidae Central Symmetry System (CSS).

The paper combines comparative morphology, gene expression analysis, genome sequencing, and RNAseq-based transcriptomics. The use of multiple approaches strengthens the conclusions and allows for a more comprehensive understanding of wing pattern evolution at the level of the CSS. The authors should be commended for generating a high-quality reference genome for *E. clarus*. This resource will likely be a valuable contribution to the field and will support future comparative genomic studies in Lepidoptera.

In addition, the authors attempted to generate WntA CRISPR knockouts but they were unsuccessful. Nonetheless they persisted in their efforts to obtain functional evidence for the role of WntA in CSS and they used heparin injections to perturb WntA signaling. The differential effects on forewing and hindwing patterns they observed support dual nature of CSS homologue in *E. clarus* (but see below for my concerns).

The authors performed RNAseq analysis across proximal, medial, and distal wing regions, which allowed them to identify key transcription factors and signaling genes (e.g., lobe, odd-paired, u-shaped, homothorax, vestigial) that may regulate pattern formation. The potential conservation of P-D patterning mechanisms between Diptera and Lepidoptera is an interesting idea/proposition.

Finally, a nice touch to the narrative of the study and something that increases the scientific impact of the work presented in the manuscript is that it revisits and validates century-old hypotheses by Schwanwitsch regarding the CSS.

1. My main criticism of the paper is the complexity of data presentation. Simplified/less cluttered figures could improve clarity.

2. The manuscript starts smoothly and easy to read in the first part of the introduction but it quickly becomes dense and highly technical, which may limit accessibility for readers outside the evo-devo or butterfly genomics communities. Maybe a within reason increase of the text to explain things in a slightly more unassuming manner would help. The authors should keep in mind as their target audience evo-devo people who are not butterfly or even insect specialists.

3. RNAseq was performed at early pupal stages (12% and 16%), which may not capture the full dynamics of gene expression during wing pattern formation. The authors could consider adding additional time points could provide a more complete developmental trajectory. But I understand that this may not be easy to do for various pragmatic reasons. At least they should explain their choice of the specific pupal stages (for readers who are not experts in the field) and discuss potential limitations of using those stages.

4. Although the CRISPR attempts were unsuccessful in the hands of the authors, the use of heparin injections to perturb WntA signaling provided (some) functional insights. The differential effects on forewing and hindwing patterns support the dual nature of CSS homologue in *E. clarus*. However, heparin can affect other signaling pathways as the authors acknowledge. I am wondering if it is possible for the authors to use a knockdown approach instead (e.g. RNAi) if this technique can be performed (i.e. the injection part) in a manner similar to the heparin injections?

5. While the authors identify several transcription factors as candidates for playing a role in CSS signaling, their direct roles in pattern formation are not experimentally validated. I am not expecting the authors to perform functional assays to assess their regulatory interactions with WntA in this manuscript but at least they can suggest follow-up experiments in the discussion.

6. In Figure 4 panels E and F there is a wealth of data that in the results section is not really described/discussed in the results section. The discussion is focused to the GO terms enrichment from panel 4D but not 4E and 4F. If I get correctly the authors basically say that 4E and 4F validate 4D. I would expect them to bring up the transcription factors and signaling molecules that are for example strongly upregulated or downregulated in 12% pupa in the FW and HW (medial section).

Figure related points:

7. Scale bars for Figure 2 A are missing.

8. Text in Figure 2B is extremely pixelated. Please replace it.

9. Scale bars are needed throughout figure 3

10. Figure 3: Please indicate in the figure the number of animals that were injected in each case and the percentage showing an expansion of the hindwing silver band and the forwing orange bands.

11. In Figure 4 panels E and F the text must be replaced since it looks fuzzy and basically it is very difficult to read it.

Reviewer 2

Comments for the author

This paper examines wing pattern evolution in Hesperidae using transcriptomic/genomic approaches and developmental genetics. In doing so, the study bridges a long-standing gap between classical morphological theory (Schwanwitsch's homology framework) and bit of modern molecular evo-devo. The authors demonstrate WntA expression as a positional marker for the Central Symmetry System (CSS) in skippers, providing evidence for developmental homology with nymphalid butterflies. The addition of an annotated *E. clarus* genome adds value to the manuscript and provides a valuable resource for future work.

Overall, I found this paper conceptually strong, data rich, well-written, and fun to read. However, some aspects of the methods require additional clarification, and I would encourage the authors to put fairly niche concepts (e.g., pierellization, CSS, Schwanwitsch's homology framework) into broader context so their work reaches a bigger pool of evolutionary biologists. This would help reduce what feels like somewhat of a burden of heavy discipline-specific jargon at various points throughout the paper. It would help to include a final 'conclusions' or 'significance' section (channelling e.g. what the authors have written in lines 349-351) that emphasizes the big-picture take-homes from this work; as is, the final 'new insights in the regulatory genetics' section gets specific very quickly, some of the significance is lost.

Finer points:

Clearly state total sample sizes (i.e., number of individuals sampled; #/sex; I don't recall seeing this in the text).

The manuscript identifies 265 P-D DEGs, but it's unclear whether DESeq2 results were corrected for multiple testing and if replicate quality was assessed.

Add details on FDR thresholds, normalization, and batch correction (if any).

Throughout, consider citing comparable findings in *Heliconius* or *Papilio* to contextualize the results (more emphasis was placed on *Drosophila*).

The inference that lobe, odd-paired, and u-shaped are involved in patterning is interesting but fairly speculative; tone down language.

The term "dual CSS homolog" is a bit confusing. Readers may interpret it as implying duplication rather than differential deployment.

Clarify terminology to reflect spatial divergence of a single homologous element rather than two independent ones.

Clarify why 12% and 16% pupal timepoints were selected; specifically address correspondence to specific developmental stages.

Specific comments:

Line 162: Clarify what is meant by "17% pupal wings".

Line 211: How were P-D sections classified for each individual's wing (assuming that there is subtle variation in wing shape and size)? Describe standardization procedures.

Lines 287-289: Clarify - the melanic shutter pattern was only apparent with heparin injections and did not appear to be the typical expressed phenotype.

Line 298: Concept of pierellization needs to be introduced.

Line 396: From where were 'non-wing tissue' transcriptomes source? i.e., which tissue? How many transcriptomes were generated from how many individuals?

Line 434: More detail needed here. The heparin injections are interpreted as WntA activation, but these manipulations are nonspecific and could affect multiple Wnt ligands or morphogens.

Lines 471-472: Which groups were compared, and was any effort made to control for DE between sexes (as is known to affect expression results)? (I do not recall the authors discussing sampling of sexes).

Figure 3: Make white arrows indicating shutter expansions a different color - the white is too hard to see, particularly against the white background.

Reviewer's Responses to Questions

Experimental quality

Does each figure have the proper controls?

If 'No', please indicate reasons in Comments for Author box below.

Reviewer #1:

- Yes

Reviewer #2:

- Yes

Were the data analyzed using appropriate statistical tests?

If 'No', please indicate reasons in Comments for Author box below.

Reviewer #1:

- Yes

Reviewer #2:

- Yes

Reproducibility

Were experiments performed using adequate number of biological replicates?

If 'No', please indicate reasons in Comments for Author box below.

Reviewer #1:

- Yes

Reviewer #2:

- Yes

Does the methods section provide sufficient detail to permit reproducibility?

If 'No', please indicate reasons in Comments for Author box below.

Reviewer #1:

- Yes

Reviewer #2:

- Yes

Completeness

Are the manuscript's conclusions supported by the data?

If 'No', please indicate reasons in Comments for Author box below.

Reviewer #1:

- Yes

Reviewer #2:

- Yes

Scholarship

Do the authors cite and discuss the merits of data that would argue for and against their conclusion?

If 'No', please indicate reasons in Comments for Author box below.

Reviewer #1:

- Yes

Reviewer #2:

- Yes

Does the manuscript title & abstract accurately reflect the contents of the manuscript, without hyperbole?

If 'No', please indicate reasons in Comments for Author box below.

Reviewer #1:

- Yes

Reviewer #2:

- Yes

First revisionAuthor response to reviewers' comments

Dear Dr. Armstrong,

We were pleased and impressed by the quality of the reviews and the clarity of your guidance. We wish to sincerely thank the reviewers for their thoughtful comments, and we used their feedback to revise our manuscript as detailed in the point-by-point responses below.
Kind regards,

Arnaud Martin and Jasmine Alqassar (on behalf of all co-authors).

Blue : Author's response ; Orange : in-text revisions (new text in bold)

Reviewer 1

The study investigates wing pattern formation in Hesperidae, a butterfly family that has been largely understudied. By focusing on *Epargyreus clarus*, the authors provide rather compelling evidence for developmental homology between the Hesperidae and Nymphalidae Central Symmetry System (CSS).

The paper combines comparative morphology, gene expression analysis, genome sequencing, and RNAseq-based transcriptomics. The use of multiple approaches strengthens the conclusions and allows for a more comprehensive understanding of wing pattern evolution at the level of the CSS. The authors should be commended for generating a high-quality reference genome for *E. clarus*. This resource will likely be a valuable contribution to the field and will support future comparative genomic studies in Lepidoptera.

In addition, the authors attempted to generate WntA CRISPR knockouts but they were unsuccessful. Nonetheless they persisted in their efforts to obtain functional evidence for the role of WntA in CSS and they used heparin injections to perturb WntA signaling. The differential effects on forewing and hindwing patterns they observed support dual nature of CSS homologue in *E. clarus* (but see below for my concerns).

The authors performed RNAseq analysis across proximal, medial, and distal wing regions, which allowed them to identify key transcription factors and signaling genes (e.g., lobe, odd-paired, u-shaped, homothorax, vestigial) that may regulate pattern formation. The potential conservation of P-D patterning mechanisms between Diptera and Lepidoptera is an interesting idea/proposition.

Finally, a nice touch to the narrative of the study and something that increases the scientific impact of the work presented in the manuscript is that it revisits and validates century-old hypotheses by Schwanwitsch regarding the CSS.

We appreciate this excellent summary of our article.

1. My main criticism of the paper is the complexity of data presentation. Simplified/less cluttered figures could improve clarity.

We agree that Figs. 1, 4 and 5 are particularly dense but opted to keep them in our revision without simplification.

Figure 1 : it is worth noting that the Snail Plot (Fig. 1B) and the Merian element painting (Fig 1C)

are also featured in all the butterfly genome notes from the Darwin Tree of Life project and Project Psyche in Lepidoptera. We believe these panels are important in that context, notably because the quality of the *E. clarus* annotation makes it an important resource for future work in skipper comparative genomics.

We did our best to summarize our transcriptomics results in Fig. 4 with an Upset Plot, and reduced heatmaps focused on two categories relevant to development. We think that developmental geneticists will recognize some gene names that should peak their interests.

2. The manuscript starts smoothly and easy to read in the first part of the introduction but it quickly becomes dense and highly technical, which may limit accessibility for readers outside the evo-devo or butterfly genomics communities. Maybe a within reason increase of the text to explain things in a slightly more unassuming manner would help. The authors should keep in mind as their target audience evodevo people who are not butterfly or even insect specialists.

We took this suggestion at heart and enriched our Introduction in several places, expanding upon general principles of butterfly wing evo-devo in the text.

3. RNAseq was performed at early pupal stages (12% and 16%), which may not capture the full dynamics of gene expression during wing pattern formation. The authors could consider adding additional time points could provide a more complete developmental trajectory. But I understand that this may not be easy to do for various pragmatic reasons. At least they should explain their choice of the specific pupal stages (for readers who are not experts in the field) and discuss potential limitations of using those stages.

As reinforced in our revised Introduction, our goal was to capture the early steps of spatial pattern formation. The Results paragraph already mentions that the 16-17% stages were chosen to match existing data about *WntA* expression in nymphalid butterflies. The fragility of wing tissues makes *in situ* hybridizations difficult prior to this stage : the wings shear and yield results that are not interpretable.

The bulk of our transcriptomic data was at the 12% stage which we now justify in the Results accordingly:

To gain preliminary insights on this process, we conducted an RNAseq analysis of pupal forewings and hindwings, each sectioned into proximal, medial, and distal portions (Fig. 4A). While *WntA* *in situ* hybridizations were conducted at around 17% of pupal development, we sampled wing transcriptomes at 12% of pupal development instead, in order to better capture early signaling events relevant to spatial pattern formation.

4. Although the CRISPR attempts were unsuccessful in the hands of the authors, the use of heparin injections to perturb *WntA* signaling provided (some) functional insights. The differential effects on forewing and hindwing patterns support the dual nature of CSS homologue in *E. clarus*. However, heparin can affect other signaling pathways as the authors acknowledge. I am wondering if it is possible for the authors to use a knockdown approach instead (e.g RNAi) if this technique can be performed (i.e. the injection part) in a manner similar to the heparin injections?

Pupal wing RNAi can be performed in some lepidopteran species by cutting the early pupal cuticle prior to hardening, peeling it back, placing a droplet with the RNAi reagents, and electroporating the wing (Ando & Fujiwara 2013). However, this method is not viable in this species as the cuticle is not malleable.

5. While the authors identify several transcription factors as candidates for playing a role in CSS signaling, their direct roles in pattern formation are not experimentally validated. I am not expecting the authors to perform functional assays to assess their regulatory interactions with *WntA* in this manuscript but at least they can suggest follow-up experiments in the discussion.

We think that our findings about the median expression (potentially correlated with WntA) of *u-shaped*, *odd-paired*, and *Lobe* warrants further investigations, perhaps in more tractable nymphalid species. We added a note on this in the last Discussion section.

Little is known so far about the expression of *u-shaped*, *odd-paired*, and *Lobe* in lepidopteran wings. **In situ hybridization experiments in skippers and nymphalids will be particularly useful to determine whether they are associated with WntA patterning, and if their expression undergoes divergent or conserved evolution.**

6. In Figure 4 panels E and F there is a wealth of data that in the results section is not really described/discussed in the results section. The discussion is focused to the GO terms enrichment from panel 4D but not 4E and 4F. If I get correctly the authors basically say that 4E and 4F validate 4D. I would expect them to bring up the transcription factors and signaling molecules that are for example strongly upregulated or downregulated in 12% pupa in the FW and HW (medial section).

The main goal of Figs. 4D-E-F is to summarize the transcriptomics data with a developmental lens, and we prefer to keep the descriptions of these results brief. In addition to the genes we further detailed in Figure 5, the last paragraph of the Discussion also mentions a few more genes of interest that appear in the Figure 4 heatmaps.

Figure related points:

7. Scale bars for Figure 2A are missing.

We do not have a proper scale reference for these images.

8. Text in Figure 2B is extremely pixelated. Please replace it.

Our revision now includes high resolution files.

9. Scale bars are needed throughout figure 3

Corrected.

10. Figure 3: Please indicate in the figure the number of animals that were injected in each case and the percentage showing an expansion of the hindwing silver band and the forewing orange bands.

We added to the legend :

Effects were replicated across several individuals : 5/5 (12.5 µg), 1/3 (20 µg), 2/2 (25 µg).

The lack of replication at the intermediate dosage could be likely explained by a failed injection or by improper staging. We had few individuals to work with in the first summer, but the strong effects we obtained at these higher dosages (20-25 µg) compelled us to focus on a lower dosage the following year.

11. In Figure 4 panels E and F the text must be replaced since it looks fuzzy and basically it is very difficult to read it.

Fixed.

Reviewer 2

This paper examines wing pattern evolution in Hesperidae using transcriptomic/genomic approaches and developmental genetics. In doing so, the study bridges a long-standing gap between classical morphological theory (Schwanwitsch's homology framework) and bit of modern molecular evo-devo. The authors demonstrate WntA expression as a positional marker for the

© 2025. Published by The Company of Biologists under the terms of the Creative Commons Attribution License (<https://creativecommons.org/licenses/by/4.0/>).

Central Symmetry System (CSS) in skippers, providing evidence for developmental homology with nymphalid butterflies. The addition of an annotated *E. clarus* genome adds value to the manuscript and provides a valuable resource for future work.

Overall, I found this paper conceptually strong, data rich, well-written, and fun to read. However, some aspects of the methods require additional clarification, and I would encourage the authors to put fairly niche concepts (e.g., pierellization, CSS, Schwanwitsch's homology framework) into broader context so their work reaches a bigger pool of evolutionary biologists. This would help reduce what feels like somewhat of a burden of heavy discipline-specific jargon at various points throughout the paper. It would help to include a final 'conclusions' or 'significance' section (channelling e.g. what the authors have written in lines 349-351) that emphasizes the big-picture take-homes from this work; as is, the final 'new insights in the regulatory genetics' section gets specific very quickly, some of the significance is lost.

We thank the reviewer for these comments and suggestions. Although we could not completely avoid jargon, we enriched the Introduction to give it a more general tone, and also added a conclusion statement that emphasizes significance.

Conclusion

Overall, this work shows that developmental signal WntA shapes butterfly wing evolution across deep time, and uncovers new candidate genes for color patterning. Moreover, the apparent similarities in wing spatial expression suggest that lepidopteran wing development may share deeply conserved mechanisms of proximo-distal patterning with *Drosophila*.

Finer points:

Clearly state total sample sizes (i.e., number of individuals sampled; #/sex; I don't recall seeing this in the text).

We added sample sizes for the heparin experiments in the legend of Figure 3.

The RNAseq sample sizes were unequal across stage and tissues but are visible in the number of columns in the heatmaps (eg. in Fig. 4E-F), and number of dots in the gene expression profiles (Fig. 5).

The methods now clearly state that mixed sexes were used. **Wing transcriptomes of both male and female individuals are available under the NCBI Bioproject PRJNA660444 [...]**
The new PCA analysis (Fig. S4) does not indicate a major effect of sex (see below).

The manuscript identifies 265 P-D DEGs, but it's unclear whether DESeq2 results were corrected for multiple testing and if replicate quality was assessed. Add details on FDR thresholds, normalization, and batch correction (if any).

We clarified the false discovery rate used after the Benjamini-Hochberg method for multiple testing correction, as implemented in DESeq2, in the Methods as well as in the following lines: **DESeq2 analysis identified 1,480 differentially expressed genes (DEGs) between the forewing and hindwing tissues (Fig. 4B-C; adjusted $p < 0.05$) [...]**

Next, we investigated the set of genes showing a pattern of differential expression along the proximo-distal axis of each wing (P-D axis DEGs). A total of 265 P-D axis DEGs were identified (Fig. 4B; **adjusted $p < 0.05$**) [...]

We also assessed replicate quality and batch effects by performing principal component analysis and verified that the samples included in the analysis clustered as expected (developmental time and wing type [e.g. forewing or hindwing]). We added the following new

lines to our methods section:

The *vst()* output was also used to run a principal component analysis with the *plotPCA()* function of DESeq2 (Fig. S4)

And we added a PCA plot as a new supplemental figure (Fig. S4).

Throughout, consider citing comparable findings in *Heliconius* or *Papilio* to contextualize the results (more emphasis was placed on *Drosophila*).

Rather than piece-mealing more similarities in the butterfly literature, we added comparative insights with more distant lineages instead in our Discussion. These parallels provide more macroevolutionary, generally applicable insights.

Our data also adds to the emerging evidence that this conservation of P-D patterning extends to other insect lineages such as Coleoptera and Orthoptera. Of note, both *sd* and *vg* show distal expressions in nymphalid butterflies (Banerjee et al., 2025; Carroll et al., 1994), and *sd* shows distal expression in the developing wings of crickets (Yamashita et al., 2023). A distal bias expression of *omb* was previously documented in butterflies, beetles and crickets (Banerjee and Monteiro, 2023; Ohde et al., 2022; Tomoyasu et al., 2009). And lastly, both *fat* and *ff* showed a distal enrichment in developing cricket wings (Ohde et al., 2022). More systematic studies of the expression and function of these genes are needed in various insect lineages, but the current data point at a deep homology of wing proximo-distal patterning across both hemimetabolous and holometabolous insects.

The inference that lobe, odd-paired, and u-shaped are involved in patterning is interesting but fairly speculative; tone down language.

We added qualifier language to the corresponding Discussion section:

Little is known so far about the expression of *u-shaped*, *odd-paired*, and *Lobe* in lepidopteran wings. **In-situ hybridization experiments in skippers and nymphalids will be particularly useful to determine whether they are associated with WntA patterning, and if their expression undergoes divergent or conserved evolution.**

The term "dual CSS homolog" is a bit confusing. Readers may interpret it as implying duplication rather than differential deployment.

We agree that dual can mean "consisting of two parts" or "two aspects or identities". We considered "bifunctional", "biform", "bipartite" and "dichotomous", but found that these terms all carry their own ambiguities. For lack of a better term, we maintain the use of dual in the sense of "consisting of two aspects or identities".

Clarify terminology to reflect spatial divergence of a single homologous element rather than two independent ones.

We enhanced in both the revised figures and text that the CSS can undergo a **dislocation along the Cu1 vein**. This should clarify this issue, by making it clear that a single homologous element can take a split appearance (in a way that was proposed and predicted by mid-20th century morphologists).

Clarify why 12% and 16% pupal timepoints were selected; specifically address correspondence to specific developmental stages.

Please refer to our reply to Reviewer 1 on this exact topic.

Specific comments:

© 2025. Published by The Company of Biologists under the terms of the Creative Commons Attribution License (<https://creativecommons.org/licenses/by/4.0/>).

Line 162: Clarify what is meant by "17% pupal wings".

We edited the manuscript in various places to emphasize that 12%, 16% and 17% each refer to a pupal stage expressed in % of total pupal development. The methods include the following: Pupal wings were dissected at 48-52 hr after pupation at 26 °C, corresponding to about 17% of pupal development (average total pupal development time at 26 °C was measured as 292 h, N =8).

Line 211: How were P-D sections classified for each individual's wing (assuming that there is subtle variation in wing shape and size)? Describe standardization procedures.

We clarified in the Methods that Fig. 4A lays out the dissection plan we used when dissecting these wings:

Each wing dissection used vein landmarks in order to result in similar sections across samples, as schematized in Fig. 4A.

Lines 287-289: Clarify - the melanic shutter pattern was only apparent with heparin injections and did not appear to be the typical expressed phenotype.

The forewing melanic shutters express WntA and match the contouring of wild-type orange stripes (ie. the typical phenotype). We made their annotation more prominent in Fig. 3E, which should make more sense now at the light of the legend and text.

Line 298: Concept of pierellization needs to be introduced.

We clarified the Discussion to allow the reader to better visualize this effect in the figures.

In nymphalids, the CSS is normally expressed as a continuous antero-posterior stripe, but it is also commonly observed in a dislocated configuration with a break or shift along the Cu1 vein, a phenomenon known as pierellization (Nijhout, 1991; Otaki, 2021; Penz, 2017; Schwanwitsch, 1925). The observed breaks of the CSS we document in the forewings of at least two skipper lineages are reminiscent of this effect (Figs. 3E and 6). Thus, the developmental mechanisms that fuel and constrain the evolution of wing patterns are similar between nymphalids, hesperiids, and beyond.

Line 396: From where were 'non-wing tissue' transcriptomes source? i.e., which tissue? How many transcriptomes were generated from how many individuals?

We added the following:

Non-wing tissue transcriptomes generated for annotation purposes— a whole third instar larvae, a fifth instar larval head, a pair of fifth instar silk glands, a male adult head, a female adult head, the testes from one adult male, and the ovaries from one adult female—were obtained by tissue storage in TRI Reagent [...]

Line 434: More detail needed here. The heparin injections are interpreted as WntA activation, but these manipulations are nonspecific and could affect multiple Wnt ligands or morphogens.

We discussed the need for nuanced interpretation in the Results section, for example:

Despite the caveat that heparin injections may affect other signaling ligands, the combination of WntA expression assays and pharmacological perturbation indicate that WntA functions as a melanic shutter in forewings and as a silver band activator in hindwings.

Lines 471-472: Which groups were compared, and was any effort made to control for DE between sexes (as is known to affect expression results)? (I do not recall the authors discussing sampling of sexes).

The sampling used mixed sexes but our new Fig. S4 does not indicate a major effect of sex, most of the variation among pupal samples stemmed from the early pupal wings.

Figure 3: Make white arrows indicating shutter expansions a different color - the white is too hard to see, particularly against the white background.

Corrected.

Second decision letter

MS ID#: bio.062297R1

MS Title: WntA expression and wing transcriptomics illuminate the evolution of stripe patterns in skipper butterflies

Authors: Jasmine D. Alqassar, Teomie S. Rivera-Miranda, Joseph J. Hanly, Christopher R. Day, Silvia M. Planas Soto-Navarro, Paul B. Frandsen, Riccardo Papa and Arnaud Martin

I am happy to tell you that your manuscript has been accepted for publication in Biology Open, pending our standard publication integrity checks. It was accepted on 28th October 2025.